# A Techno-Economic Feasibility Analysis of Mono-Si and Poly-Si Photovoltaic Systems in the Rooftop Area of Commercial Building under the Feed-In Tariff Scheme

**Ke Shi [1,2], Chuangyi Li [3] and Choongwan Koo [4,\*]**

[1] National Institute of Clean-and-Low-Carbon Energy, Beijing 102211, China; ke.shi.h@chnenergy.com.cn
[2] Center for Green Energy and Architecture, China Energy Investment Corporation, Beijing 102211, China
[3] The Country Garden, Guangzhou 510000, China; lichuangyiup@163.com
[4] Division of Architecture & Urban Design, Incheon National University, Incheon 22012, Korea
\* Correspondence: cwkoo@inu.ac.kr

**Abstract:** Hong Kong's government has recently introduced the feed-in tariff scheme to promote the photovoltaic (PV) system as a promising way to address global warming. The feed-in tariff scheme depends on the type of the PV system and its installed capacity. This study aimed to investigate the techno-economic feasibility of mono-Si and poly-Si PV systems in the rooftop area of a commercial building, Pao Yue-Kong Library of Hong Kong, under the feed-in tariff scheme. The analysis was carried out in two phases: (i) technical analysis of the rooftop PV systems by considering the shading effect and solar radiation and (ii) economic feasibility of the rooftop PV systems under the feed-in tariff scheme from the life cycle perspective. The main findings of the case study can be summarized: (i) the rooftop area of the target building would not be significantly affected by surrounding buildings; (ii) the highest amount of solar radiation was estimated at 136.96 kWh/m$^2$ in October, while the lowest value was 55.64 kWh/m$^2$ in February; (iii) the total amount of module energy yield from the mono-Si PV system was estimated at 917.58 kWh/kW, indicating that it was very similar but a little bit lower (i.e., 0.48%) than that for the poly-Si PV system (i.e., 921.98 kWh/kW); and (iv) payback periods for mono-Si and poly-Si PV systems were estimated at 8.67 and 8.31 years, respectively. The feasibility study can contribute to providing facility managers with a practical guideline to determine the appropriate strategy in implementing the PV systems in buildings under the feed-in tariff scheme.

**Keywords:** mono-Si photovoltaic system; poly-Si photovoltaic system; shading effect; energy generation; techno-economic feasibility; feed-in tariff

## 1. Introduction

### 1.1. Research Background

Since global warming has become a worldwide issue, it was emphasized in the Paris Agreement in December 2015 that the increase in the global average temperature should be limited to 2 °C above the pre-industrial levels [1]. The decarbonization of the conventional energy system is considered an effective way to cope with the global warming problem. In this aspect, renewable energy resources can play a crucial role in reducing the use of fossil fuels and realizing energy system transformation [2].

Renewable energy resources (e.g., solar, wind, geothermal, biomass, etc.) will never run out, and will not emit greenhouse gases or atmospheric pollutants. Among various options, the solar photovoltaic (PV) system, which directly converts sunlight into electricity, is the most widely used approach because of its safe, reliable, efficient characteristics. In the building sector, the interest in building-integrated photovoltaic (BIPV) systems is consistently increasing, and it is expected to improve the energy self-sufficiency rate of buildings by implementing the PV system on its rooftop and façade areas [3,4].

In metropolitan cities such as Hong Kong with high building density and limited land area, it is generally recommended to install the PV system on building surface areas [5]. Since Hong Kong has no natural energy resources [6], it is required to reduce its dependence on imported energy resources by implementing local renewable energy resources. In particular, the solar radiation potentials are abundant in Hong Kong, which is much higher than that in European countries [7,8]. Despite the suitability of implementing the PV system in Hong Kong, the installed PV power capacity is still limited, being less than 6.29 MWp by March 2017 [9]. The Hong Kong government has taken considerable efforts to encourage the application of solar resource [10]. In particular, the feed-in tariff scheme for renewable energy systems was launched in October 2018 [11], and it had brought more than 7000 applications of the PV system at the end of 2019 [12].

### 1.2. Literature Review

With the theme of the implementation of the PV system in buildings, various previous studies have been conducted to evaluate the technical performance of the PV system and its economic feasibility. First of all, it is essential to understand the impact factors of the PV system in terms of the technical performance of the PV system. Koo et al. [13] categorized the impact factors of the PV system into two groups. The first category included the regional climate factors, such as latitude, monthly meridian altitude, monthly average daily solar radiation, and monthly average temperature. The other category included the building characteristics and the physical features of the PV system, such as the azimuth and slope of the installed panel, the type of panel and inverter, and the rooftop area. In particular, the efficiency of the PV panel depends on the type of the PV panel, largely affecting the power output of the PV system. Among the three generations of PV systems, the first generation has been widely used, and the second and third generations share a small percentage in the market. The first generation mainly uses silicon wafers, including mono-crystalline silicon (mono-Si) and poly-crystalline silicon (poly-Si). Agrawal et al. [14] introduced the efficiency of different PV panels, with the results showing that the PV system with crystalline silicon had higher efficiency than others. Yang et al. [8] analyzed the output performance of mono-Si, poly-Si, and others, indicating that the mono-Si performed the best in terms of energy output per square meter.

Second, some studies estimated the annual energy harvesting of PV systems. Rus-Casas et al. [15] classified the calculation methods of energy harvesting in two groups, i.e., indirect methods and direct methods. The indirect methods first calculate the power capacity and then calculate the energy generation, while the direct methods directly calculate the energy generation. The indirect methods can be divided into three types: (i) methods that generate the I–V curve of a generator based on the equivalent circuit models [16,17] (i.e., one-diode model and two-diodes model) of the solar cell or based on an artificial neural network (ANN) [18]; (ii) methods that translate known I–V curves at a given operating condition to the desired operating conditions [19–22], in which some impact factors such as series resistance, temperature coefficients, and solar irradiance are considered; (iii) methods that obtain the power from some atmospheric parameters and from information provided by manufacturers [16,23]. Since the indirect methods based on I–V curves need additional information such as indoor measurements and outdoor measurements, they are complex and more difficult to use than direct methods. The direct methods that define the behavior of the PV generator include four methods: (i) the method based on the average annual radiation on the PV plane, the conversion efficiency, the generator area, and the installed power [16]; (ii) the method based on the DC performance ratio [24,25]; (iii) the method based on liner regression models that require radiation and temperature data [26,27]; and (iv) the method based on the energy rating characterization of the modules in the generator [28,29].

Meanwhile, other studies considered the economic aspects for the PV system, such as the renewable energy policy and economic feasibility from the life cycle perspective. First, according to the development progress of PV technology in some countries such as

mainland China, EU, and the US, it is required to promote the PV system with a financial support from the government so as to improve the price competitiveness of electricity generated from the PV system. Many previous studies reported that the government's incentive policy could have an essential influence on the PV market. Namely, the PV industry could be rapidly developed with sufficient financial support from the government, while it could decrease if the government reduced the financial support. In Japan, the government subsidy program in 1994 significantly lowered the PV installation cost, leading to a notable increase in the PV market [30,31]. Then, the PV subsidy was terminated during 2005–2008; accordingly, the installed capacity decreased from 290 MW to 225 MW. Thereafter, the Japanese government re-introduced economic policy (i.e., the feed-in tariff) in 2009 to motivate the PV market, and the newly installed capacity was more than double that of 2008 [32]. In China, the government initiated various incentive policies such as the renewable energy law, rooftop subsidy program, national feed-in tariff scheme, and the golden sun program. The feed-in tariff scheme (introduced in 2011) provided an RMB 1/kWh tariff to the owners of the PV system, leading to a dramatic increase in the Chinese PV capacity. The newly installed capacity reached 5000 MW in 2012 [33,34]. Second, many previous studies analyzed the economic feasibility and profitability of the PV system. Koo et al. [13] conducted the life cycle cost analysis of the PV system in South Korea by considering several factors such as regional climate factors and building characteristics, resulting in an optimal implementation strategy for the PV system. Bakos et al. [35] analyzed the economic performance of the BIPV system under the different percentage of government subsidy, indicating that the promotion of PV application should be initially enforced by the government. Additionally, the result showed a long-term payback period without the government subsidy. Peng and Lu [5] and Li et al. [36] compared the economic performance of the PV system between Hong Kong and other countries, with results that showed the PV installation cost in Hong Kong was much higher than that in PV leading countries such as mainland China and Germany. Lang et al. [37] assessed the economic feasibility of the rooftop PV system in Europe, and results showed that electricity prices were of primary importance, and self-consumption for rooftop PV was economically attractive for many buildings in central Europe, even without subsidies. Beuse et al. [38] investigated the profitability of PV projects in South-East Asia, finding that profitable investment opportunities exist in commercial and industrial PV projects. Zhao et al. [39] investigated the economic performance of the rooftop PV system in China, with results showing that the payback period was only 4.0–8.0 years, and the average net present value was 6.45 CNY/W. Lou et al. [40] analyzed the economic aspect of the rooftop PV system in Hong Kong, indicating that it would take 9–16.8 years to recover the initial investments by the benefits.

### 1.3. Challenges and Objectives

As mentioned above, previous studies have analyzed the technical performance of the PV system in buildings by considering the various impact factors; however, they rarely sought to compare the techno-economic feasibility between mono-Si and poly-Si PV systems in buildings by considering the feed-in tariff scheme as the government's financial support as well as the technical impact factors such as the layout of PV panels, the shading effect from surrounding buildings, and solar radiation. To address these challenges, this study aimed to investigate the techno-economic feasibility of mono-Si and poly-Si PV systems in the rooftop area of commercial buildings (i.e., an educational facility located in Hong Kong) under the feed-in tariff scheme. The type of educational facility generally has a sufficient rooftop area so as to install a large amount of the PV panels. The analysis process can be explained in two phases: (i) technical analysis of the rooftop PV systems (by considering the reference model, shading effect, solar radiation potential, and electricity generation) and (ii) economic feasibility of the rooftop PV systems under the feed-in tariff scheme (by considering the net present value and the saving-to-investment ratio from the life cycle perspective). This study can contribute to providing facility managers

with a practical guideline to investigate the technical and economic feasibility of different types of the PV systems under the feed-in tariff scheme in implementing a renewable and sustainable energy systems in buildings.

## 2. Materials and Methods

The techno-economic feasibility analysis of the rooftop PV system in this study was conducted in two phases. As shown in Figure 1, for Phase 1, a technical analysis of the rooftop PV system was carried out to estimate the amount of solar radiation and electricity generation by considering the shading effect and the specifications of the PV systems. Then, for Phase 2, the economic feasibility of the rooftop PV system was evaluated based on the results obtained in Phase 1 as well as all of the costs and benefits of the PV system from the life cycle perspective.

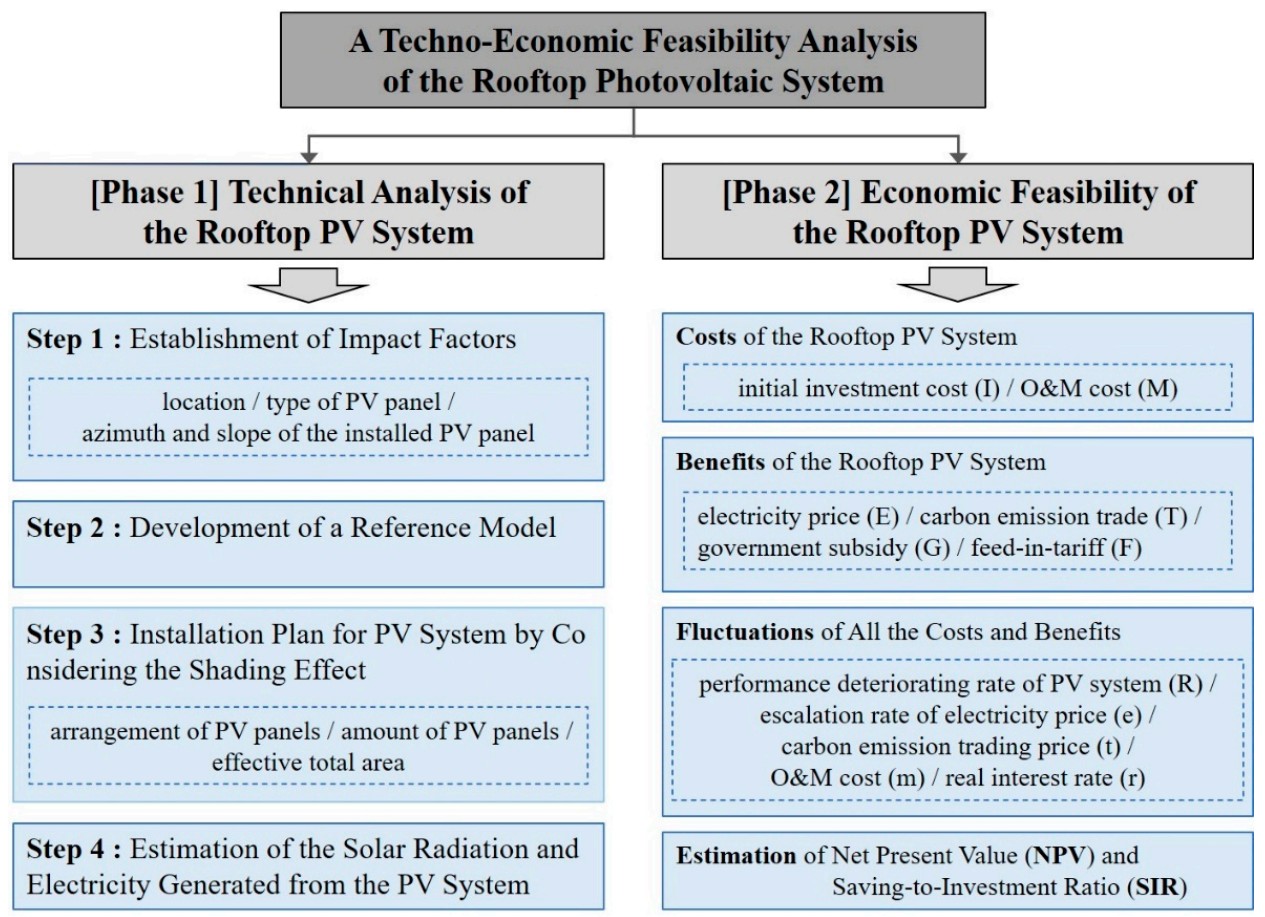

**Figure 1.** Research framework for the techno-economic feasibility of the rooftop PV system.

### 2.1. Phase 1: Technical Analysis of the Rooftop PV System

This study analyzed the technical performance between mono-Si and poly-Si PV systems in the rooftop area under Hong Kong climate conditions, which was conducted in three steps: (i) the establishment of impact factors; (ii) the development of a reference model; (iii) an installation plan for the PV system by considering the shading effect; and (iv) the estimation of the amount of solar radiation and electricity generated from the PV system.

### 2.1.1. Establishment of Impact Factors

Based on the previous studies, the impact factors can be mainly classified into three aspects: (i) location; (ii) type of PV panel; and (iii) azimuth and slope of the installed PV panel.

- Location: Hong Kong is located in the subtropical area with an abundant solar radiation resource, of which the longitude and latitude are $114°15'$ E and $22°15'$ N. Peng and Lu [5] compared the solar radiation resource between Hong Kong and some PV leading countries and regions. It was shown that the annual solar radiation for mainland China, Germany, and the US were 1000–2200, 900–1200 and 1200–2400 $kWh/m^2$, respectively, while that for Hong Kong was determined to be 1333 $kWh/m^2$. Thus, it can be said that the solar radiation potentials in Hong Kong are sufficient enough to implement the PV system in buildings.
- Type of PV panel: The efficiency of the PV panel depends on the type of the PV panel, which could largely affect the power output of the PV system. The PV technology is generally classified into three generations (refer to Table 1): (i) the first generation mainly uses silicon wafers, including mono-Si and poly-Si; (ii) the second generation involves thin-film technologies, including amorphous silicon (a-Si), cadmium telluride (CdTe), and copper indium gallium selenide (CIGS); and (iii) the third generation covers organic, dye-sensitized solar panels, perovskite, etc. [8,41,42].

**Table 1.** Generation of PV system and its efficiency.

| Generation of PV System and Its Materials | | Efficiency (%) |
|---|---|---|
| First generation (crystalline silicon) | Monocrystalline (mono-Si) | 15–21% |
| | Polycrystalline (poly-Si) | 14–18% |
| Second generation (thin film) | Amorphous silicon (a-Si) | 6–9% |
| | Cadmium telluride (CdTe) | 14–17% |
| | Copper indium gallium selenide (CIGS) | 10–15% |

- Azimuth and slope of the installed PV panel: Al-Otaibi et al. [43] analyzed the azimuth and slope of the installed PV panel, indicating that the azimuth of the PV panel should face the south in the northern hemisphere, and it should be opposite in the southern hemisphere. In terms of the slope of the PV panel, it usually is equal to the latitude of the actual location of a PV project so that more solar radiation can reach the PV panel. Peng and Lu [5] found the optimal slope of the installed PV panel in Hong Kong, indicating that the optimal tilt angle would be $22 \pm 1$ degree close to the local latitude.

### 2.1.2. Development of a Reference Model

It is required to establish a reference model for the Pao Yue-Kong Library ($114.19°$ E, $22.30°$ N) as a target building so as to analyze the technical performance of the rooftop PV system in the target building. First, Google Earth, a virtual Earth software that places satellite and aerial photos, and the relevant physical information on a 3D model of the Earth, was used to collect the layout and physical information of the target buildings. Second, based on the collected information on the target buildings, Ecotect, software for building energy simulation, was used to build the three-dimensional reference model.

Figure 2 shows the layout image of the Pao Yue-Kong Library and the surrounding buildings captured from *Google Earth*. It was shown that block M, block DE and block EF were higher than the Pao Yue-Kong Library, indicating that the shading effect caused by block M, block DE and block EF should be considered in carrying out the technical analysis. Thus, four buildings (i.e., Pao Yue-Kong Library, Block M, Block DE, and Block EF) were considered in developing a reference model, in which the physical information on these buildings could be calculated by using the *three-dimension measurement* function in *Google Earth*. Finally, a reference model was built via *Ecotect* based on the actual size and location of the target buildings (refer to Figure 3).

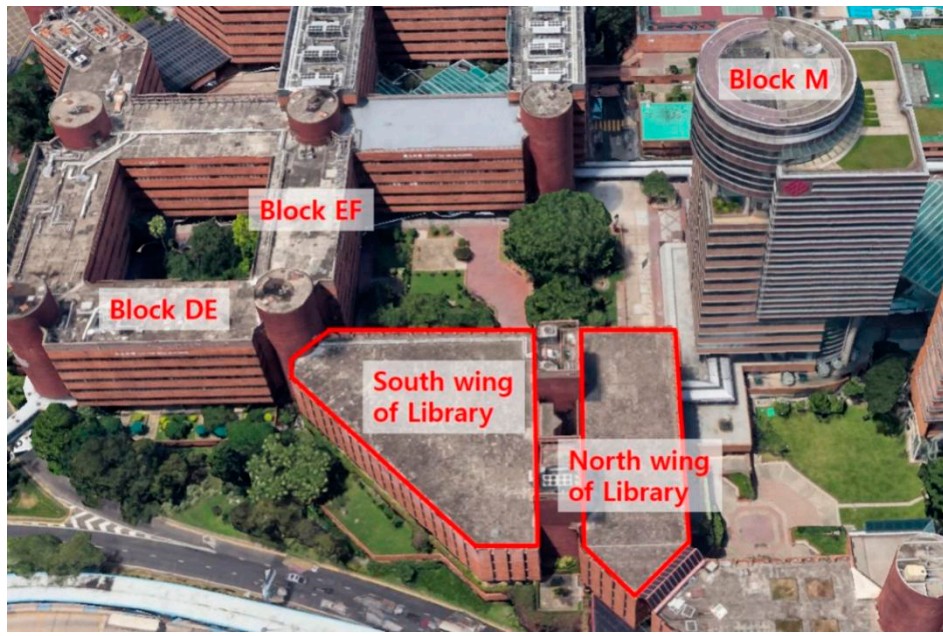

**Figure 2.** Pao Yue-Kong Library and surrounding buildings in Google Earth.

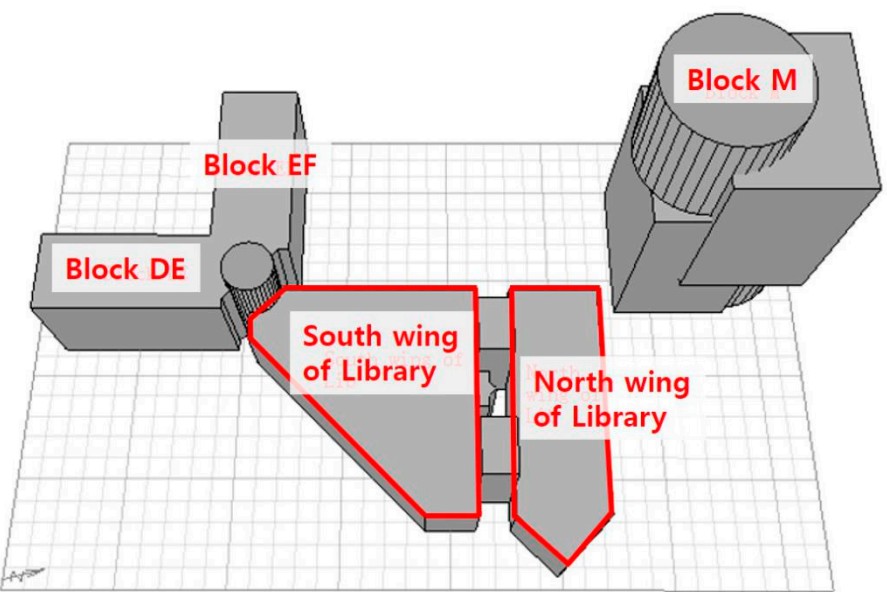

**Figure 3.** Reference model for analyzing the technical performance of PV system.

2.1.3. Installation Plan for PV System by Considering the Shading Effect

The amount of solar radiation is one of the most important factors that can affect the amount of electricity generated from the PV system. Meanwhile, the shadows from the surrounding buildings can have a negative effect on the amount of solar radiation reached on the PV system, especially in high-rise and high-density cities such as Hong Kong. Since the surrounding buildings (i.e., block M, block DE and block EF) are higher than the target building (i.e., Pao Yue-Kong Library), it is essential to analyze the shading effect of these buildings before estimating the amount of electricity generated from the rooftop PV system. Ecotect, a software program, was used to simulate the shading effect from the surrounding buildings on the target building by considering the relevant information on location (i.e., latitude, altitude), orientation, and weather data. Considering the sun path of a day is similar within the season [44], four representative days in each season (i.e., spring equinox, summer solstice, autumn equinox, and winter solstice) were selected to analyze the shading

effect from the surrounding buildings. Supplementary Material (SM) Figures S1–S4 show the shading profiles on the target building on the typical days of four seasons.

As explained in Equation (1), the shaded area can be estimated by considering the percentage of shading that can be determined via the Shading and Shadow function in Ecotect, in which the operating time from 9:00 a.m. to 5:00 p.m. on the typical days of four seasons were considered.

$$\text{Shaded area} = \text{Total rooftop area} \times \text{percentage of shading} \tag{1}$$

where, a total of rooftop area of the Pao Yue-Kong Library is 3021.75 m$^2$.

In addition to the shading effect from the surrounding buildings, the shading from the front row of the PV panels needs to be considered as well. If the adjacent two rows of the PV panels are too close, the shadow from the front row of the PV panels could have an effect on the rear row of the PV panels. Thus, it is required to carefully calculate the shading distance between two rows of the PV panels so as to avoid the shading from the front row of the PV panels. The shading distance of the PV panels can be calculated using Equations (2) and (3) (refer to Figure 4).

$$A = L \times \frac{sin\alpha}{tan\beta} \tag{2}$$

$$\beta = 90^{\circ} - \varphi - \varepsilon \tag{3}$$

where, $A$ stands for the shading distance of the PV panels; $L$ stands for the length of the PV panel; $\alpha$ stands for the installed angle of the PV panel; $\beta$ stands for the meridian altitude at noon of winter solstice (i.e., the value for Hong Kong is 44.2°); $\varphi$ stands for the regional latitude; and $\varepsilon$ stands for the tilt angle of the Earth's axis (i.e., the value for Hong Kong is 23.5°) [45].

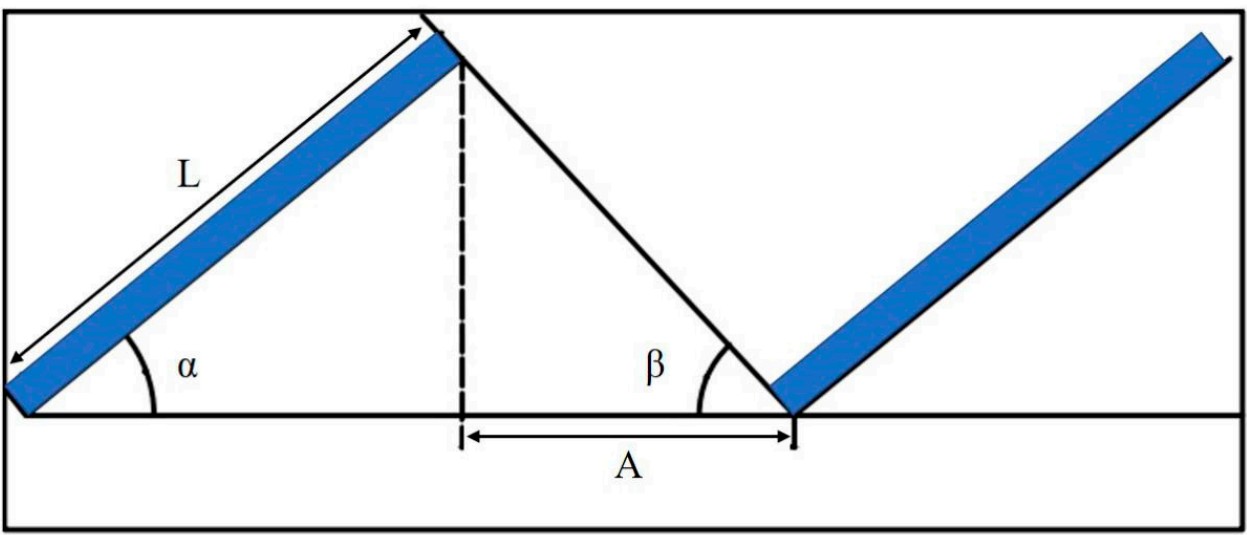

**Figure 4.** Spacing between the PV panels due to the shading effect.

### 2.1.4. Estimation of the Amount of Solar Radiation and Electricity Generated from the PV System

This study compared the technical performance between mono-Si and poly-Si PV systems in the rooftop area of the target building. As shown in Table 2, the installed PV panels were produced from Csun Solar Tech, of which the size (length × width) was 1.96 m × 0.99 m. For the mono-Si PV panel, the capacity per unit and efficiency were found to be 0.355 kW and 18.30%, respectively; and for the poly-Si PV panel, 0.310 kW and 16.00%.

**Table 2.** Physical information on two types of the PV systems.

| Classification | Mono-Si PV System | Poly-Si PV System |
|---|---|---|
| Manufacturer | Csun Solar Tech | Csun Solar Tech |
| Specification | CSUN355-72M | CSUN310-72P |
| Capacity per unit (kW) | 0.355 | 0.310 |
| Efficiency | 18.30% | 16.00% |
| Length (m) | 1.96 | 1.96 |
| Width (m) | 0.99 | 0.99 |

Table 3 shows the installation information on two types of the PV systems. The tilt angle of the PV panels was set at 22.3° by considering the latitude of Hong Kong, and the PV panels was set to be orientated towards the south. The shading distance between the adjacent two rows of the PV panels was calculated at 0.76 m via Equations (2) and (3). As shown in Figure 5, the installation scenario of the PV system in the rooftop area of the target building was determined by considering the PV panel's size, tilt angle, orientation, shading effects from surrounding buildings, shading space, and other factors. Due to this, the PV panels would be generally installed towards the south, while the target building would not be facing south, so the PV panels could not be arranged along the edge of rooftop. In addition, there were some maintenance space, equipment space and space between the north wing and the south wing of the library's rooftop area. Accordingly, the effective total area, including the projection of PV panels area and the shading space area, was determined to be 2117 m$^2$ (70% of the total rooftop area). For both mono-Si and poly-Si PV systems, a total of 832 PV panels was determined to be installed in the rooftop area of the Pao Yue-Kong Library, and the installed capacity of two types of the PV systems were calculated to be 295.36 kW and 257.92 kW, respectively.

**Table 3.** Installation information on two types of the PV systems.

| Classification | Mono-Si PV System | Poly-Si PV System |
|---|---|---|
| Tilt angle | 22.3° | 22.3° |
| Orientation | South | South |
| Shading space | 0.76 m | 0.76 m |
| Number of panels | 832 | 832 |
| Installed capacity | 295.36 kW | 257.92 kW |

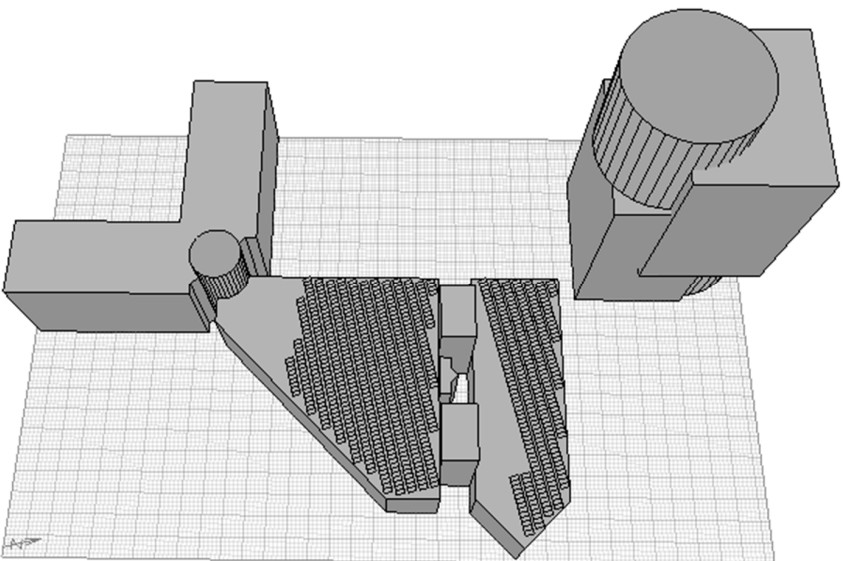

**Figure 5.** Installation scenario of the PV system in the rooftop area of Pao Yue-Kong Library.

As shown in Supplementary Materials Figures S5–S40, the shading effects were analyzed from 9:00 to 17:00 on the typical days of four seasons (i.e., spring equinox, summer solstice, autumn equinox, and winter solstice). Although some areas would be shaded from 9:00 to 17:00 on the winter solstice, it was determined to install the PV panels in all of the rooftop area because the other seasons with more solar radiation were rarely shaded. For example, as shown in Figures 6–9, the shading percentage at 15:00 of the winter solstice was found to be 20% while the values for the spring equinox, summer solstice, and autumn equinox were shown to be 4%, 0% and 4%, respectively. Thus, almost all of the Pao Yue-Kong Library's rooftop area would be used for installing PV panels in the spring, autumn, and summer season without considering the shading effects from surrounding buildings, but not in the winter season.

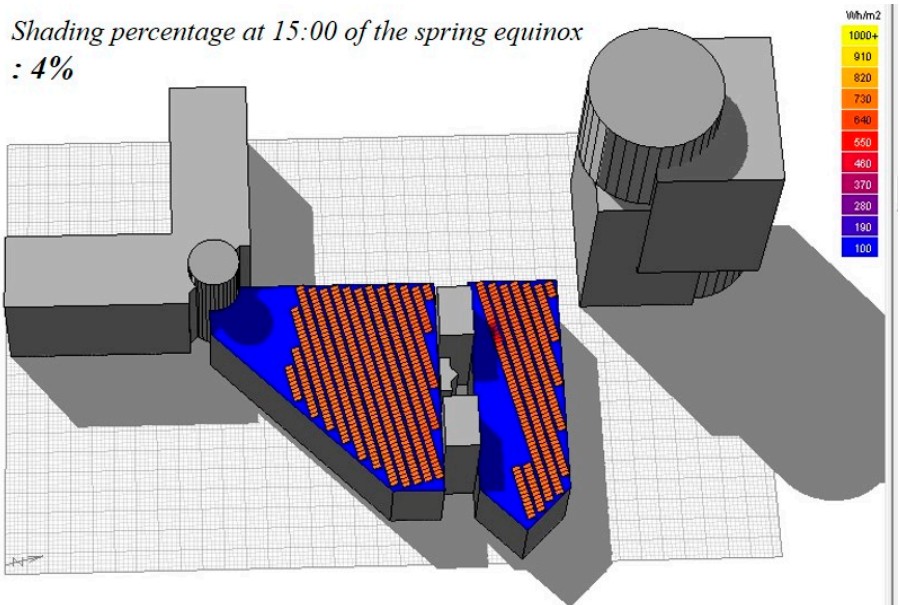

**Figure 6.** Shading effect on the target building at 15:00 p.m. of the spring equinox.

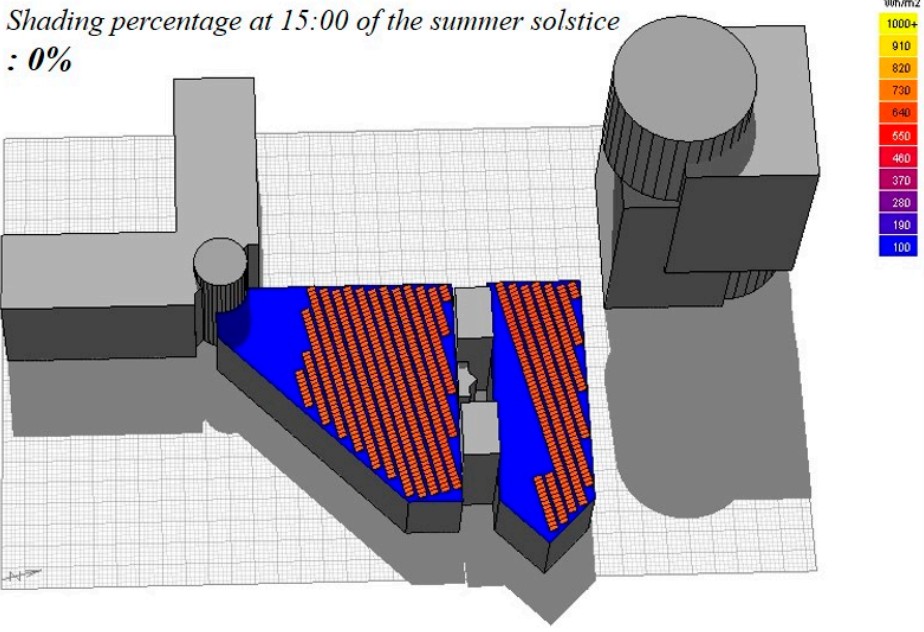

**Figure 7.** Shading effect on the target building at 15:00 p.m. of the summer solstice.

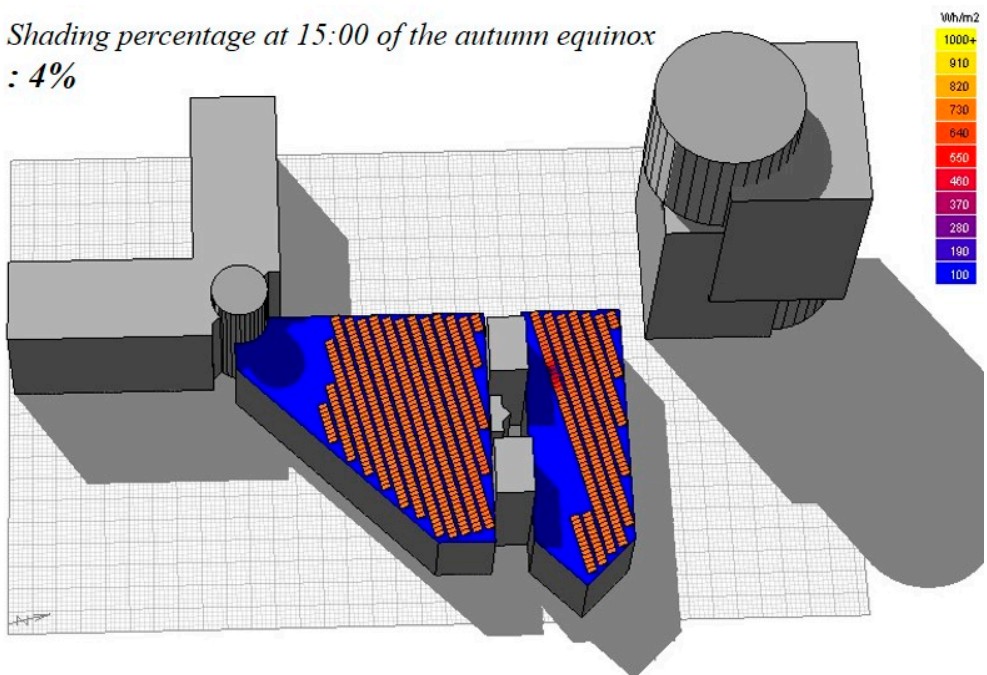

**Figure 8.** Shading effect on the target building at 15:00 p.m. of the autumn equinox.

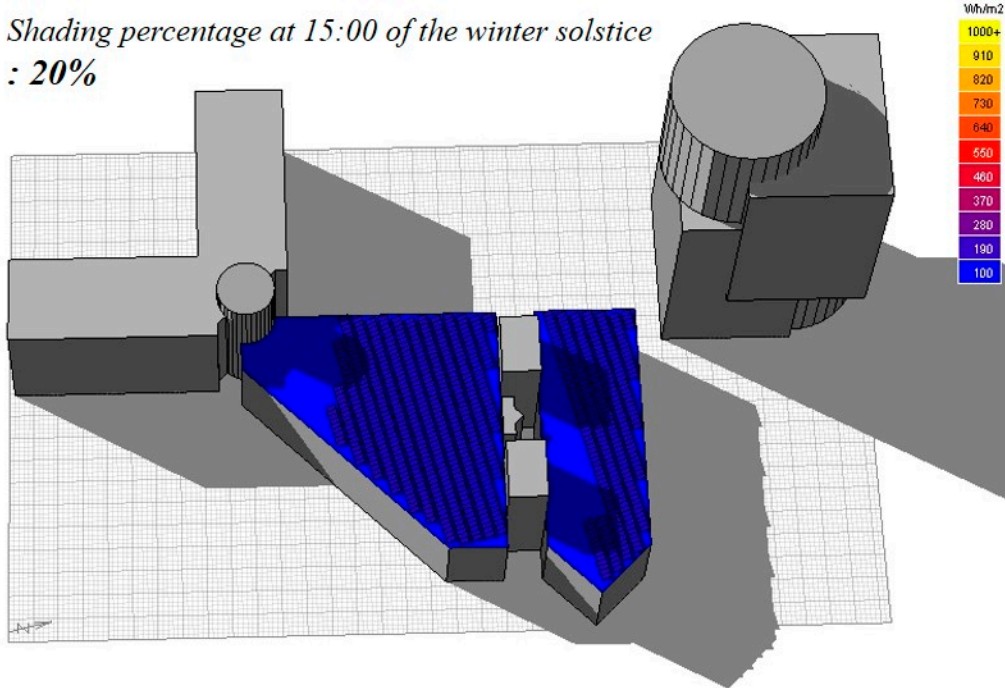

**Figure 9.** Shading effect on the target building at 15:00 p.m. of the winter solstice.

Based on the installation scenario of the PV systems, the monthly average solar radiation reached on the PV panels could be estimated using the Solar Access Analysis function in Ecotect. The total solar radiation was calculated by Equation (4).

$$\text{Total solar radiation} = \text{Average solar radiarion} \times \text{Exposed areas} \qquad (4)$$

where a total of the exposed area of the PV panels in the rooftop area of the Pao Yue-Kong Library was calculated via Ecotect, which was 1614.43 m$^2$.

Finally, the amount of electricity generated from the rooftop PV system in the Pao Yue-Kong Library could be calculated by the energy simulation in Ecotect. Since the settings of the energy simulation in the software program include the type of PV panels, the efficiency of PV panels, the area of PV panels, and the total solar radiation reached on the PV plane, the energy simulation method of the software was presumed to use the direct methods (refer to Section 1.2). In order to obtain a value close to the real electricity generation of PV system, it is required to consider some important impact factors such as the outdoor temperature, increase in surface temperature of the PV panel, the series resistance and other miscellaneous loss so as to correct the simulation results. Since it is complex to comprehensively consider these impact factors in the calculation formula as well as difficult to obtain them, this study employed a performance ratio, $\lambda$, to correct the results of electricity generation obtained by energy simulation. The electricity generation can be estimated via Equation (5).

$$E = E_0 \times \lambda \tag{5}$$

where $E$ is the electricity generation; $E_0$ is the electricity generation obtained by energy simulation; $\lambda$ is the performance ratio of the PV system, which was set at 0.75 in this study.

### 2.2. Phase 2: Economic Feasibility of the Rooftop PV System

The life cycle cost analysis (LCCA) is one of the traditional methods used to evaluate the economic feasibility of a new technology in the building sector. This study used the LCCA to compare the economic feasibility between mono-Si and poly-Si PV systems to be applied to the rooftop area of the Pao Yue-Kong Library. In general, the LCCA includes two main indexes, i.e., the net present value (*NPV*) as an absolute index and the saving-to-investment ratio (*SIR*) as a relative index. The present worth method was adopted, and the analysis period was set at 25 years by considering the life span of the PV systems.

As shown in Equations (6)–(17), the *NPV* and *SIR* can be calculated by considering all the costs (i.e., initial investment and operation and maintenance) and benefits (i.e., electricity price, carbon emission trading, government subsidy, and feed-in tariff) for the PV systems. If '$NPV \geq 0$' or '$SIR \geq 1$', the PV systems would be regarded as profitable.

$$NPV = \sum_{i=0}^{n} \frac{E_i + T_i + G_i + F_i}{(1+r)^i} - \sum_{i=0}^{n} \frac{I_i + M_i}{(1+r)^i} \tag{6}$$

$$SIR = \sum_{i=0}^{n} \frac{E_i + T_i + G_i + F_i}{(1+r)^i} \Big/ \sum_{i=0}^{n} \frac{I_i + M_i}{(1+r)^i} \tag{7}$$

$$E_i = E_0(1-R)^i(1+e)^i \tag{8}$$

$$T_i = T_0(1-R)^i(1+t)^i \tag{9}$$

$$G_i = G_0(1-R)^i(1+g)^i \tag{10}$$

$$F_i = F_0(1-R)^i \tag{11}$$

$$I_i = I_0 \tag{12}$$

$$M_i = M_0(1+m)^i \tag{13}$$

where *NPV* is the net present value; *SIR* is the saving-to-investment ratio; $n$ is the lifespan of the PV system (25 years was determined in this study); $i$ is the system operation time in years; $r$ is the real discount rate; $E_t$ is the benefit from electricity price; $T_t$ is the benefit from carbon emission trading price; $G_t$ is the benefit from government subsidy price; $F_t$ is the benefit from feed-in tariff price; $I_t$ is the initial investment cost; $M_t$ is the operation and maintenance (O&M) cost; $R$ is the performance degradation rate of the PV systems; $e$ is the

escalation rate of electricity price; $t$ is the escalation rate of carbon emission trading price; $g$ is the escalation rate of government subsidy; and $m$ is the escalation rate of O&M cost.

$$E_0 = EG \times P_E \tag{14}$$

$$T_0 = EG \times P_T \times f \tag{15}$$

$$G_0 = EG \times P_G \tag{16}$$

$$F_0 = EG \times P_F \tag{17}$$

where EG is the electricity generation (kWh); $P_E$ is the electricity price (USD/kWh); $P_T$ is the carbon emission trading price (USD/kg); $f$ is the carbon emission factor (kg/kWh); $P_G$ is the government subsidy price (USD/kWh); and $P_F$ is the feed-in tariff price (USD/kWh).

### 2.2.1. Costs of the Rooftop PV System

Table 4 shows the cost information of two types of PV systems, which mainly includes initial investment cost and O&M cost. The initial investment includes the costs for the PV module, inverter, other hardware, and labor. According to the previous materials [6,33], the initial investment cost ranged from USD 2504/kW through to USD 5600/kW during 2012–2020. Based on the open data [46], published by the China Light and Power company (CLP), one of the power companies in Hong Kong, the latest initial investment cost of a PV system ranged from USD 3120/kW through to USD 5200/kW. In this study, two types of PV systems (i.e., CSUN355-72M for mono-Si and CSUN310-72P for poly-Si) and the relevant materials were determined. The initial investment costs were estimated at USD 3354/kW for the mono-Si PV system and USD 3250/kW for the poly-Si PV system. Additionally, the O&M cost was determined to be 0.5% of the initial investment cost for each year [47]. The HKD–USD exchange rate was used at 1.00:0.13, referring to HSBC [48].

**Table 4.** Cost information on two types of the PV system.

| Classification | Mono-Si PV System | Poly-Si PV System |
|---|---|---|
| Unit cost per capacity | USD 3354/kW | USD 3250/kW |
| Initial investment cost for PV system ($I_0$) | USD 990,637 | USD 838,240 |
| O&M cost ($M_0$) | 0.5% of the initial investment cost for each year | 0.5% of the initial investment cost for each year |

As mentioned in the previous section (Phase 1: technical analysis), the total of the installed capacity of the mono-Si PV system was estimated at 295.36 kW; and the poly-Si PV system, 257.92 kW. Thus, the initial investment costs for two types of PV systems were determined at USD 990,637 and USD 838,240, respectively.

### 2.2.2. Benefits of the Rooftop PV System

The benefits from the installed PV systems could be determined by considering the electricity price, carbon emission trading price, government subsidy, and feed-in tariff. Hong Kong's government introduced the feed-in tariff scheme for renewable energy systems in October 2018, including the PV system and wind power system [49]. Under this scheme, all the amount of electricity generated from the PV system can be sold to the power company until 2033, and the project owner can gain the benefits from the feed-in tariff. After 2033, the amount of electricity generated from the PV system will be owned and consumed by the project owner, and the project owner can gain the benefits from the electricity savings.

As shown in Table 5, the feed-in tariff scheme can be classified into three levels depending on the capacity of the installed PV systems [49]. As mentioned above, two types of the PV systems were assumed to be installed at 295.36 kW and 257.92 kW, respectively; and thus, the feed-in tariff was determined to be USD 0.39/kWh. In addition, the electricity

price was set at USD 0.1541/kWh, according to the CLP [50]. Furthermore, the carbon emission trading price was set at USD 0.0041/kg, referring to the Guangzhou carbon market [51,52], and the carbon emission factor was set at 0.62 kg/kWh from the CLP [53]. Meanwhile, the government subsidy was eliminated from Equation (6) as Hong Kong's government currently does not have the subsidy policy. Table 6 shows the benefits from the installed PV systems. The HKD–USD and HKD–CNY exchange rates were used at 1.00:0.13 and 1.00:0.90 from HSBC [48].

**Table 5.** Feed-in tariff for the PV systems in Hong Kong.

| Capacity of the Installed PV Systems | Feed-in Tariff ($P_F$) |
|---|---|
| ≤10 kW | USD 0.65/kWh |
| 10 kW–200 kW | USD 0.52/kWh |
| 200 kW–1 MW | USD 0.39/kWh |

**Table 6.** Benefits from the installed PV systems.

| Variable | Description | Value |
|---|---|---|
| $P_E$ | Electricity price | USD 0.1541/kWh |
| $P_T$ | Carbon emission trading price | USD 0.0041/kg |
| $f$ | Carbon emission factor | 0.62 kg/kWh |
| $P_F$ | Feed-in tariff | USD 0.39/kWh |

### 2.2.3. Fluctuations of all the Costs and Benefits from the PV Systems in Hong Kong

Table 7 shows the fluctuations of all the costs and benefits from the PV system in Hong Kong, which should be considered in the LCCA. The performance degradation rate of the PV systems was set at 0.8%/year. The annual escalation rate of electricity price was estimated at 3.0%/year based on the average value of the past ten years from the CLP [32] (refer to Table 8). The escalation rate of the carbon emission trading price and O&M cost were set at 3.0% [54] and 1.0% [40], respectively. In addition, the real interest rate was determined by considering the nominal interest rate and inflation rate (refer to Equation (18)). The nominal interest rate was assumed at 2.0%, according to the Hong Kong Monetary Authority [55]. The inflation rate was estimated at 3.3% based on the calculation of the average annual escalation rate of the Consumer Price Index in Hong Kong for the past decade (2010–2019) (refer to Table 9) [56]. Thus, the real interest rate was estimated at −1.3%.

$$r = \frac{1 + r_n}{1 + r_{inf}} - 1 \tag{18}$$

where r stands for the real interest rate; $r_n$ stands for the nominal interest rate; and $r_{inf}$ stands for the inflation rate.

**Table 7.** Fluctuations of all the cost and benefits from the PV systems in Hong Kong.

| Variables | Description | Value |
|---|---|---|
| R | Performance degradation rate of the PV systems | 0.8%/year |
| e | Escalation rate of electricity price | 3.0% |
| t | Escalation rate of carbon emission trading price | 3.0% |
| m | Escalation rate of O&M cost | 1%/year |
| r | Real interest rate | −1.3% |

**Table 8.** Electricity price and its escalation rate in Hong Kong (2010–2019).

| Year | 2010 | 2011 | 2012 | 2013 | 2014 | 2015 | 2016 | 2017 | 2018 | 2019 |
|------|------|------|------|------|------|------|------|------|------|------|
| Electricity price ($P_E$) | 91.6 | 94.2 | 98.7 | 104.7 | 111.0 | 110.3 | 113.2 | 110.5 | 115.4 | 118.5 |
| Escalation rate (e) | 2.8 | 2.8 | 4.8 | 6.1 | 6.0 | 0.3 | 2.6 | −2.4 | 4.2 | 2.6 |
| Average escalation rate (e) | 3.0 | | | | | | | | | |

Note: unit—electricity price (HKD/kWh) and escalation rate (%).

**Table 9.** Escalation rate of Consumer Price Index in Hong Kong (2010–2019).

| Year | 2010 | 2011 | 2012 | 2013 | 2014 | 2015 | 2016 | 2017 | 2018 | 2019 |
|------|------|------|------|------|------|------|------|------|------|------|
| Escalation rate ($r_{inf}$) | 2.4 | 5.3 | 4.1 | 4.3 | 4.4 | 3.0 | 2.4 | 1.5 | 2.4 | 2.9 |
| Average escalation rate ($r_{inf}$) | 3.3 | | | | | | | | | |

Note: Unit—%.

## 3. Results and Discussion

This study analyzed the technical performance and the relevant economic feasibility of two types of PV systems in a commercial building (i.e., Pao Yue-Kong Library in Hong Kong). The results can be explained in the following three aspects: (i) analysis of the shading effect; (ii) estimation of the amount of solar radiation and electricity generated from the PV system; and (iii) economic feasibility of the rooftop PV system.

### 3.1. Analysis of the Shading Effect

The shading effect from the surrounding buildings was analyzed on the typical days of four seasons (i.e., spring equinox, summer solstice, autumn equinox, and winter solstice). In particular, the percentage of shading was estimated via the *Shading and Shadow* function in *Ecotect* (refer to Supplementary Materials Tables S1 and S4), with which the shaded rooftop area was determined.

Figure 10 shows the percentage of shading during the operating time from 9:00 a.m. to 5:00 p.m. on the four typical days, indicating that the shading effect would significantly occur in winter rather than in other seasons. The percentage of shading during daytime for the spring equinox, summer solstice, and autumn equinox were all lower than 16%, while that during most daytime in the winter solstice was more than 20%. In particular, the shading effect from the surrounding buildings was shown to be even more than 50% after 16:00 in the winter solstice.

Meanwhile, despite a high percentage of shading after 16:00 in the winter solstice, the rooftop area of the Pao Yue-Kong Library would not be significantly affected by the shadows from the surrounding buildings in a year. This is because the university campus was generally built with low-density buildings compared to other areas with high-density and high-rise buildings in Hong Kong. Thus, it can be concluded that educational facilities are recommended to promote the PV systems in Hong Kong.

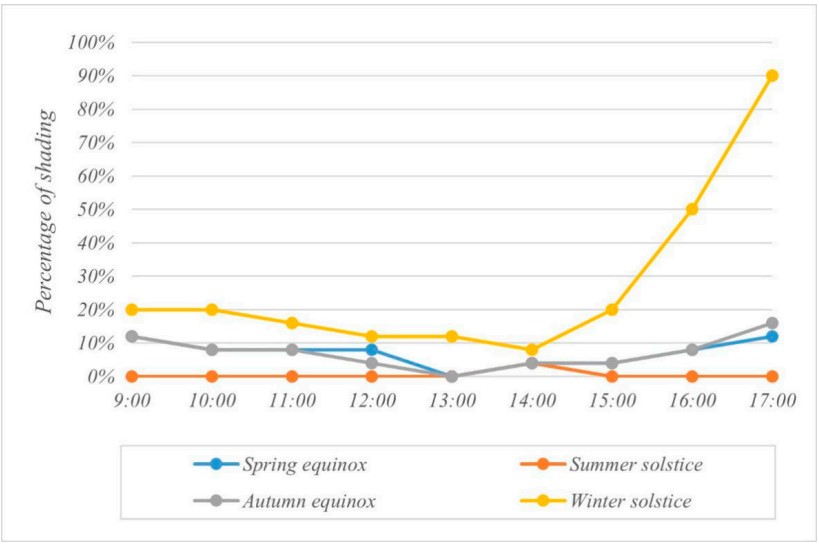

**Figure 10.** Percentage of shading on four typical days.

*3.2. Estimation of the Amount of Solar Radiation and Electricity Generated from the PV System*

Considering the shading effect from the surrounding buildings, this study analyzed the amount of solar radiation reached on the PV panels to be installed in the rooftop area of the Pao Yue-Kong Library. The solar radiation data in this study were obtained from the dataset developed by the City University of Hong Kong, in which the solar radiation from the observatory station at 22.32° N, 114.17° E on the typical meteorological year (TMY) was collected. As shown in Table 10 and Supplementary Materials Figure S41, the solar radiation was shown to be fluctuated throughout a year, which was estimated to be relatively high from May to October and low from November to April. The highest amount of solar radiation in a year was shown to be 136.96 kWh/m$^2$ in October, while the lowest value was 55.64 kWh/m$^2$ in February. A total of the monthly average solar radiation per unit area reached on the PV panels was estimated at 1223.18 kWh/m$^2$. These data were slightly lower than the largest average solar radiation calculated by Yang based on the Perez model (1333 kWh/m$^2$) [5] and the average solar radiation data of the Hong Kong Observatory between 1981 and 2020 (1302.83 kWh/m$^2$) [57].

**Table 10.** The amount of monthly solar radiation reached on the PV panels.

| Month | Monthly Average Solar Radiation per Unit Area (kWh/m$^2$) | Monthly Total Solar Radiation (kWh) |
|---|---|---|
| January | 69.80 | 112,691.46 |
| February | 55.64 | 89,822.74 |
| March | 104.53 | 168,758.28 |
| April | 71.67 | 115,710.32 |
| May | 113.84 | 183,790.64 |
| June | 100.95 | 162,971.74 |
| July | 133.04 | 214,783.25 |
| August | 130.69 | 210,996.51 |
| September | 105.88 | 170,941.07 |
| October | 136.96 | 221,116.64 |
| November | 91.62 | 147,905.86 |
| December | 108.55 | 175,251.66 |
| Total | 1223.18 | 1,974,741.38 |

Based on the simulation results of electricity generation and the correction formula in consideration of the performance ratio ($\lambda$) mentioned in Section 2.1.4, the electricity generations of two types of PV systems were obtained. Figure 11 presents the monthly

total solar radiation and the monthly total electricity generation per unit area (kWh/m$^2$); the amount of electricity generation per unit area from two types of PV systems were found to be proportional to the amount of solar radiation. This is because the amount of solar radiation reached on the PV panels should play a decisive role in the electricity generation. Table 11 shows the monthly amount of electricity generated from two types of the PV systems (i.e., mono-Si and poly-Si). The most values of the electricity generation from two types of the PV systems were shown to be higher than 50 kWh/kW throughout a year. First, in terms of the module energy yield (kWh/kW), the highest values occurred in October (i.e., 103.12 kWh/kW for mono-Si and 103.24 kWh/kW for poly-Si), while the lowest values occurred in February (i.e., 41.80 kWh/kW for mono-Si and 41.86 kWh/kW for poly-Si). As a result, a total amount of module energy yield from the mono-Si PV system was estimated at 917.58 kWh/kW, and it was determined to be very similar but a little bit lower (i.e., 0.48%) than that for the poly-Si PV system (i.e., 921.98 kWh/kW). Second, in terms of the energy yield per unit area (kWh/m$^2$), the total amount of electricity generated from the mono-Si PV system was estimated at 167.87 kWh/m$^2$, and it was determined to be 13.97% higher than that for the poly-Si PV system (i.e., 147.29 kWh/m$^2$). Compared with the previous studies, the amount of electricity generation of the PV systems in this study was similar to that in other studies of Hong Kong (i.e., 160 kWh/m$^2$ for mono-Si and 150 kWh/m$^2$ for poly-Si [5], 143 kWh/m$^2$ and 122 kWh/m$^2$ [40]).

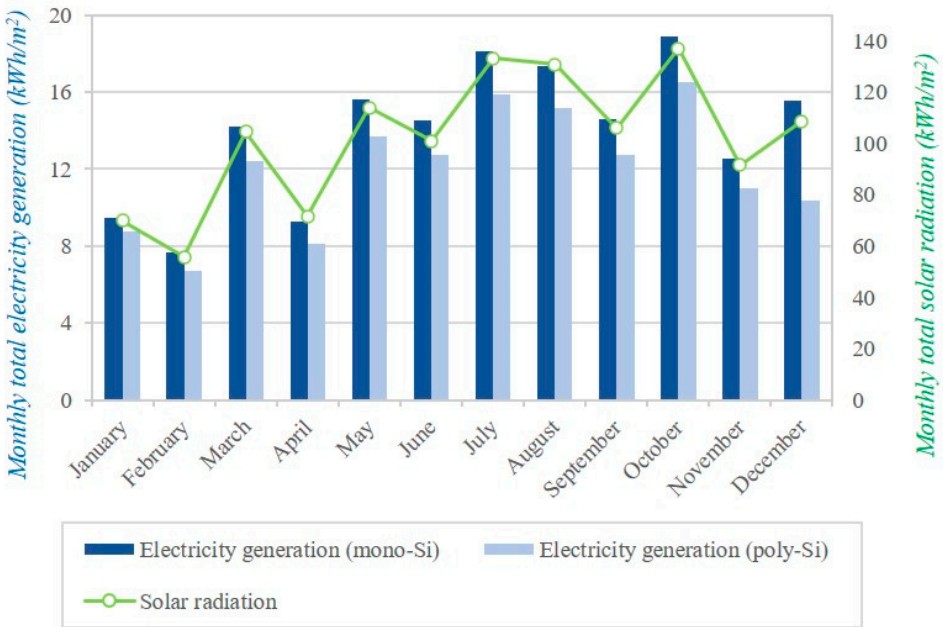

**Figure 11.** The amount of electricity generation per unit area from two types of the PV systems.

**Table 11.** The amount of electricity generated from two types of the PV systems.

| Month | The Amount of Electricity Generation | | | | | |
|---|---|---|---|---|---|---|
| | Mono-Si | | | Poly-Si | | |
| | kWh | kWh/kW | kWh/m$^2$ | kWh | kWh/kW | kWh/m$^2$ |
| January | 15,270.10 | 51.70 | 9.46 | 14,181.76 | 54.99 | 8.78 |
| February | 12,345.76 | 41.80 | 7.65 | 10,796.60 | 41.86 | 6.69 |
| March | 22,952.67 | 77.71 | 14.22 | 20,069.22 | 77.81 | 12.43 |
| April | 15,000.36 | 50.79 | 9.29 | 13,115.24 | 50.85 | 8.12 |
| May | 25,247.78 | 85.48 | 15.64 | 22,073.88 | 85.58 | 13.67 |
| June | 23,507.53 | 79.59 | 14.56 | 20,553.00 | 79.69 | 12.73 |
| July | 29,300.42 | 99.20 | 18.15 | 25,622.71 | 99.34 | 15.87 |
| August | 27,954.77 | 94.65 | 17.32 | 24,440.72 | 94.76 | 15.14 |
| September | 23,559.02 | 79.76 | 14.59 | 20,600.28 | 79.87 | 12.76 |
| October | 30,457.65 | 103.12 | 18.87 | 26,627.16 | 103.24 | 16.49 |
| November | 20,284.96 | 68.68 | 12.56 | 17,739.18 | 68.78 | 10.99 |
| December | 25,135.55 | 85.10 | 15.57 | 21,976.55 | 65.21 | 10.36 |
| **Total** | 271,016.54 | 917.58 | 167.87 | 237,796.30 | 921.98 | 147.29 |

### 3.3. Economic Feasibility of the Rooftop PV System

Considering the amount of electricity generated from two types of the PV systems (i.e., mono-Si and poly-Si PV systems to be installed in the Pao Yue-Kong Library), the economic feasibility for two PV systems was analyzed from the life cycle perspective with consideration of all the costs and benefits (refer to Section 2.2). The initial investment cost, O&M cost, electricity tariff benefit, carbon emission trade benefit and feed-in tariff benefit of the two PV systems in the first analysis year (year of 2020) was calculated, and then the values in each year during the system operation period (2020–2044) were calculated based on the first year's data and the fluctuations of all the costs and benefits (refer to Section 2.2.3). The cumulative values of costs and benefits were obtained according to the values in each year. Last but not least, the *NPV* and *SIR* were calculated based on Equations (6) and (7).

Figures 12 and 13 show the *NPV* (per unit kW) and the *SIR* for two PV systems, and Supplementary Materials Tables S5–S6 show the detailed values for the LCCA. Compared to the *NPV* (in 2044) for the mono-Si PV system (i.e., 4188 USD/kW), and the *NPV* (in 2044) for the poly-Si PV system was determined to be a little bit higher (i.e., 4348 USD/kW). In addition, the SIRs (in 2044) for the mono-Si and poly-Si PV systems were found to be 2.07 and 2.15, respectively, indicating over double benefits from the initial investment costs. It is worth noting that the *NPV* of the rooftop PV systems in Hong Kong would be significantly higher than that in other countries and regions. This is because Hong Kong's government offers a competitive feed-in tariff scheme (USD 0.39–USD 0.65/kWh) for the PV systems from October 2018.

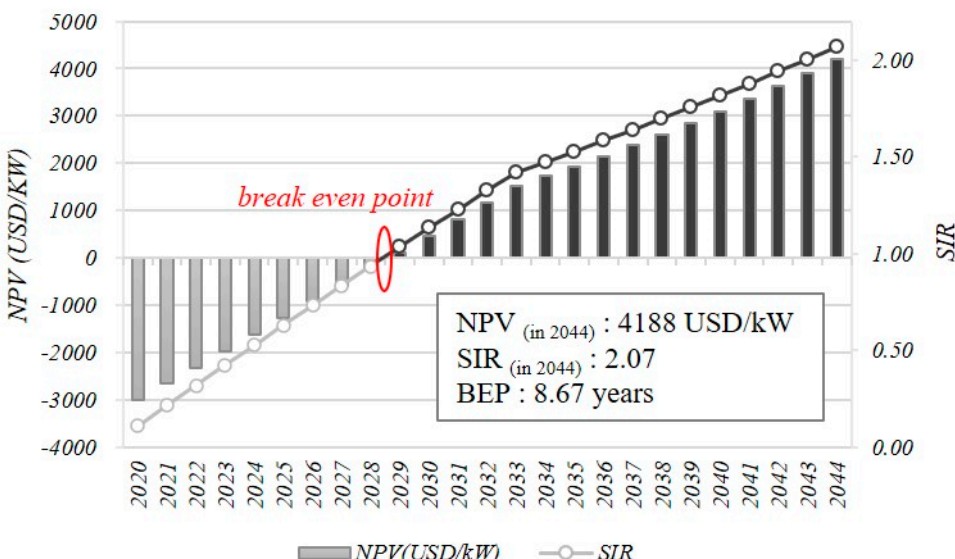

**Figure 12.** *NPV* and *SIR* for the mono-Si PV system.

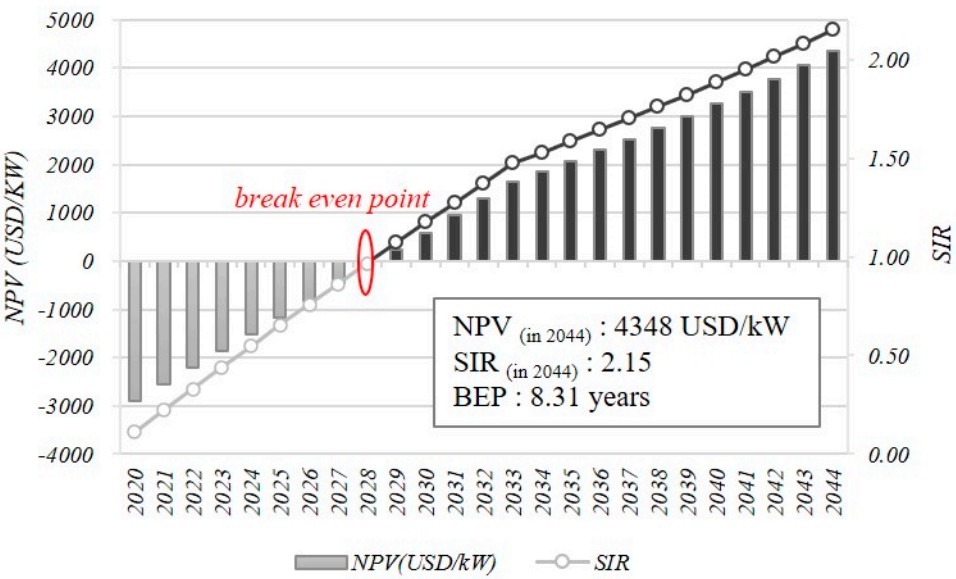

**Figure 13.** *NPV* and *SIR* for the poly-Si PV system.

Meanwhile, the payback period of two PV systems was obtained by applying '*NPV* = 0'. The payback period of the mono-Si PV system was estimated at 8.67 years, and it was determined to be 0.36 years longer than that of the poly-Si PV system (i.e., 8.31 years). It was determined that both mono-Si and poly-Si PV systems could gain profits for more than 16 years from the life span of the PV system (i.e., 25 years). The payback period of two PV systems in this study was found to be lower than that from the previous studies in Hong Kong (9–16.8 years [40], 15 years [58], and 61.4–72.4 years [43]). The result of the payback period was approximately as same as mainland China, and it was much shorter than that in many other countries and regions [59,60]. This is because Hong Kong's government offers a competitive feed-in tariff scheme (USD 0.39–USD 0.65/kWh) for the PV systems, which could promote the payback period to a relatively low level despite the high labor and other costs in Hong Kong.

## 4. Conclusions

This study aimed to investigate the techno-economic feasibility of mono-Si and poly-Si PV systems in the rooftop area of a commercial building (i.e., the Pao Yue-Kong Library located in Hong Kong) under the feed-in tariff scheme. The analysis approach was designed in two phases: (i) technical analysis of the rooftop PV systems, which was conducted in three steps (i.e., the establishment of a reference model, an installation plan for the PV system by considering the shading effect, and an estimation of the amount of solar radiation and electricity generated from the PV system) and (ii) the economic feasibility of the rooftop PV systems under the feed-in tariff scheme, by considering the *NPV* and *SIR* from the life cycle perspective. The main findings could be summarized in detail as follows.

- Shading effect: Even if the percentage of shading was found to be high in the winter solstice, it was determined that the rooftop area of the Pao Yue-Kong Library would not be significantly affected by the shadows from the surrounding buildings in a year.
- Solar radiation potential: The solar radiation reached on the PV systems in the rooftop area of the Pao Yue-Kong Library was found to be fluctuated throughout a year, showing that the highest amount of the solar radiation in a year was estimated at 136.96 kWh/m$^2$ in October, while the lowest value appeared to be 55.64 kWh/m$^2$ in February.
- Electricity generation: It was found that the amount of electricity generated from two types of PV systems (i.e., mono-Si and poly-Si) would be proportional to the amount of solar radiation, showing the highest value in October and the lowest value in February. A total amount of electricity generated from the mono-Si PV system was estimated to be 167.87 kWh/m$^2$, indicating 13.97% higher than that for the poly-Si PV system (i.e., 147.29 kWh/m$^2$).
- Economic feasibility: The *NPV* (in 2044) for the poly-Si PV system was determined to be a little bit higher (i.e., 4188 USD/kW) than that for the mono-Si PV system (i.e., 4348 USD/kW). The SIRs (in 2044) for the mono-Si and poly-Si PV systems were found to be 2.07 and 2.15, respectively, indicating over double benefits from the initial investment costs. In addition, the payback periods for the mono-Si and poly-Si PV systems were estimated at 8.67 years and 8.31 years, respectively, indicating that all the PV systems could gain profits for more than 16 years out of 25 years (the life span of the PV systems).

The results of the technical analysis and the relevant economic feasibility analysis can be used as a practical guideline for facility managers to determine the appropriate strategy in implementing the PV systems in commercial buildings that would not be severely affected by shadows (i.e., primary school, middle school, public library buildings, etc.) under the feed-in tariff scheme.

**Supplementary Materials:** The following are available online at https://www.mdpi.com/article/10.3390/su13094709/s1, Tables S1–S4 for the percentage of shading by season, Tables S5–S6 for the yearly cumulative values of costs and benefits for two types of PV systems, Figures S1–S4 for the shading profile on the target building by season, Figures S5–S40 for the shading effect on the target building by season, Figure S41 for the monthly average solar radiation reached on the PV panels.

**Author Contributions:** Conceptualization: C.L. and C.K.; Methodology: K.S. and C.K.; Validation: K.S. and C.K.; Formal Analysis: C.L.; Investigation: K.S.; Resources: K.S. and C.L.; Data Curation: K.S. and C.L.; Writing–Original Draft Preparation: K.S. and C.L.; Writing–Review & Editing: K.S. and C.K.; Visualization: K.S. and C.K.; Supervision: C.K.; Project Administration: C.K.; Funding Acquisition: C.K. All authors have read and agreed to the published version of the manuscript.

**Funding:** This work was supported by Incheon National University (International Cooperative) Research Grant in 2019 (No.2019-0355).

**Institutional Review Board Statement:** Not applicable.

**Informed Consent Statement:** Not applicable.

**Data Availability Statement:** The data presented in this study are available on request from the corresponding author.

**Conflicts of Interest:** The authors declare no conflict of interest.

## Nomenclature

**Abbreviations**

| | |
|---|---|
| a-Si | amorphous silicon |
| BIPV | building-integrated photovoltaic |
| CdTe | cadmium telluride |
| CIGS | copper indium gallium selenide |
| CLP | china light and power company |
| LCCA | life cycle cost analysis |
| mono-Si | monocrystalline |
| *NPV* | net present value |
| O&M | operation and maintenance |
| poly-Si | polycrystalline |
| PV system | photovoltaic system |
| PF | performance ratio |
| *SIR* | saving-to-investment ratio |

**Symbols**

| | |
|---|---|
| $t$ | escalation rate of carbon emission trading price |
| $e$ | escalation rate of electricity price |
| $E_i$ | benefit from electricity price |
| $f$ | carbon emission factor |
| $g$ | escalation rate of government subsidy |
| $m$ | escalation rate of O&M cost |
| $n$ | lifespan of the PV system |
| $r$ | real interest rate |
| $F_i$ | benefit from feed-in tariff price |
| $G_i$ | benefit from government subsidy price |
| EG | electricity generation |
| $R$ | performance degradation rate of the PV systems |
| $I_i$ | initial investment cost |
| $M_i$ | operation and maintenance (O&M) cost |
| $P_E$ | electricity price |
| $P_F$ | feed-in tariff price |
| $P_G$ | government subsidy price |
| $P_T$ | carbon emission trading price |
| $T_i$ | benefit from carbon emission trading price |

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
