# Peer review of "A Techno-Economic Feasibility Analysis of Mono-Si and Poly-Si Photovoltaic Systems in the Rooftop Area of Commercial Building under the Feed-In Tariff Scheme"

_sustainability, doi:10.3390/su13094709_

Round 1
Reviewer 1 Report
- Line 15: The commercial building used in this study is Pao Yue-Kong library, which shall be described in the Abstract.
- Line 21-23: (iii) a total amount of electricity generated from mono-Si PV system was estimated at 361,355.39kWh, indicating 13.97% higher than that for poly-Si PV system (i.e., 317,061.7kWh). The installed capacity of the mono-Si PV system is higher than that of poly-Si PV system. The total amount of generated electricity shall be divided by the installed capacity (kWh/kW) to demonstrate the effective efficiency.
- The longitude and latitude of Pao Yue-Kong library shall be described.
- The meanings of Figures 3-6 are similar with Figures 8-11. Figures 3-6 could be deleted, and use Figures 8-11 to demonstrate the shading effect, and describe the shading percentage at 15:00 in each figure.
- Line 208: The total of rooftop area of the Pao Yue-Kong library is 3,021.75 m2. However, this total rooftop area is not equal to the effective total area of the installed solar panels (832 panels) plus the area of shading space. The effective total area (832 panels + shading space area) shall be described in the text.
- The shading on the panels will influence the generation of electricity. The equation (1) uses the total rooftop area to evaluate the shaded area could not reveal the shading effect for evaluate the loss of the generation power of installed panels.
- Table 10 shows the amount of monthly solar radiation reached on the PV panels. The solar radiation reached on the PV panels depends on the historical local sun radiation data. It shall be described how to evaluate the historical local sun radiation data.
- Table 11 shows the amount of electricity generated from two types of the PV systems. The amount of electricity generated from the PV systems depends on the local sun radiation, and the temperature of environment and the temperature raise of the solar panel. The evaluation method to obtain close to real output of the power generation shall be described.
- The shading effect on the Pao Yue-Kong library roof is not severe. How to apply this condition to the other places in Hong Kong?
- The symbols of the calculation results of equations (5)-(17) shall be shown in the related Tables.
- The first word in the title of Table 6-9 is disappear.
Author Response
I would like to thank the reviewers for their comments and suggestions for improving the paper. Careful attention has been given to incorporate the suggestions made by the reviewers. The following sections list the major modifications that have been made in response to the reviewer comments.

Reviewer 2 Report
I think it is a very interesting piece of work, but I think it needs to be completed:
Please complete the literature review in the introduction including studies showing energy calculation methods of photovoltaic systems. I have to tell you that there are studies that make a classification of methods of estimating for annual energy harvesting calculations obtained from photovoltaic systems.
It should also complete the bibliographic review of the aspect related to the profitability of photovoltaic systems
Please incorporate a flow chart in which the methodology carried out in the work can be shown in a summarised form.
The shading simulation is very clear, but it does not apply any energy calculation method for the systems.
Please explain in detail how you obtain figure 14 and explain in an appropriate way the values obtained. Has the applied methodology been used in other studies?
The radiation data, please explain how they have been obtained.
Figures 15 and 16 are not of good quality. It would also be useful to explain how they have been obtained.
Author Response
Dear Dr. Getu Hailu:
Please find the revised manuscript of the paper entitled "A techno-economic feasibility analysis on mono-Si and poly-Si photovoltaic systems in the rooftop area of commercial building under the feed-in tariff scheme " by Shi, Li, and Koo. I would like to thank the reviewers for their comments and suggestions for improving the paper. Careful attention has been given to incorporate the suggestions made by the reviewers. The following sections list the major modifications that have been made in response to the reviewer comments.

Round 2
Reviewer 1 Report
The authors revised the manuscript and responded for the comments properly.
It could be accepted for publication in this Journal.
Reviewer 2 Report
The work has been considerably remodeled, I hope the authors are happy with the results. I congratulate the authors.